# Influence of Tethered Ions on Electric Polarization and Electrorheological Property of Polymerized Ionic Liquids

**DOI:** 10.3390/molecules25122896

**Published:** 2020-06-23

**Authors:** Fang He, Bo Wang, Jia Zhao, Xiaopeng Zhao, Jianbo Yin

**Affiliations:** Smart Materials Laboratory, Department of Applied Physics, Northwestern Polytechnical University, Xi’an 710129, China; xgdhefang@mail.nwpu.edu.cn (F.H.); wangbo0304@iccas.ac.cn (B.W.); zhaojia11@mail.nwpu.edu.cn (J.Z.); xpzhao@nwpu.edu.cn (X.Z.)

**Keywords:** polymerized ionic liquids, tethered ions, electrorheology, dielectric spectra

## Abstract

Polymerized ionic liquids (PILs) show potential to be used as new water-free polyelectrolyte-based electrorheological (ER) material. To direct ER material design at the molecular level, unveiling structure-property relationships is essential. While a few studies compare the mobile ions in PILs there is still a limited understanding of how the structure of tethered counterions on backbone influences ER property. In this study, three PILs with same mobile anions but different tethered countercations (e.g., poly(dimethyldiallylammonium) P[DADMA]^+^, poly(benzylethyl) trimethylammonium P[VBTMA]^+^, and poly(1-ethyl-4-vinylimidazolium hexafluorophosphate) P[C_2_VIm]^+^) are prepared and the influence of tethered countercations on the ER property of PILs is investigated. It shows that among these PILs, P[DADMA]^+^ PILs have the strongest ER property and P[C_2_VIm]^+^ PILs have the weakest one. By combining dielectric spectra analysis with DFT calculation and activation energy measurement, it can clarify that the influence of tethered counterions on ER property is mainly associated with ion-pair interaction energy that is affecting ionic conductivity and interfacial polarization induced by ion motion. P[DADMA]^+^ has the smallest ion-pair interaction energy with mobile ions, which can result in the highest ionic conductivity and the fastest interfacial polarization rate for its strongest ER property.

## 1. Introduction

Polymerized ionic liquids (PILs) have attracted great attention as a new type of solid polyelectrolyte in various applications from electrolytes for energy storages to electroactive components for smart materials because PILs possess not only mechanical stability of polymer and ionic conductivity of ionic liquids but also designable molecular structure or morphology [1]. For example, solid gas-sensing materials based on tetraalkylammonium-based PILs have been demonstrated to show more sensitive CO_2_ sensing behavior compared to liquid gas-sensing materials based on ILs [2]. By combing PILs with temperature-sensitive poly(N-isopropylacrylamide), soft actuators having pH and thermal dual-responsive character and high mechanical properties have been obtained [3]. By introducing gradient porous morphology into PIL membrane, high-speed solvent-responsive actuators have been developed [4]. The imidazolium sulfonate PIL composite has also been prepared and fabricated into thermally stable electroactive actuators which exhibit an effective actuation response under a low applied electrical potential of 4 V [5].

Recent study of employing hydrophobic PIL solid particles containing polyatomic fluorinated ion pairs as dispersed phase has attracted interest to develop new anhydrous electrorheological fluid (ERF) [6,7], a smart suspension whose viscosity can be controlled by electric fields owing to electric polarization and particle-particle interaction [8]. This electric field-induced viscosity thickening of ERF can be found many potential applications such as semi-adaptive damper, valve, isolator, haptic sensor, and so on [9,10,11]. Different from conventional polyelectrolyte particles whose ER property needs to activate by absorbing moisture, the dry PIL particles have strong ER property. This is because, compared to the conventional polyelectrolytes, the ion-pair interaction in PIL particles is weaker due to large and delocalized nature of polyatomic fluorinated constituent ions and, thus, the untethered ions are easy to dissociate and move to induce interfacial polarization [12]. Furthermore, the hydrophobic nature of fluorinated constituent ions also makes PIL-based ERF to be insensitive to moisture. As a result, the relationship between ER property and PIL structure can be unveiled. In particular, the number, size, and type of mobile untethered ions have been demonstrated to significantly influence the ER property of PILs and the PILs containing mobile ions with small size and large plasticization effects have strong ER property because of enhanced interfacial polarization [13,14,15,16]. In addition, we have also found that the transport of mobile ions, interfacial polarization, and ER effect can be controlled by adjusting crosslinking degree or the length of substituent alkyl chain on pedant groups [17,18]. While a few studies compare different types of mobile untethered ions, there is still a limited understanding of how the tethered counterions attached to backbone influences ER property. 

For that purpose, we herein synthesized three PILs with the same mobile untethered anions (hydrophobic hexafluorophosphate (PF_6_^−^)) but different tethered countercations attached to backbone (e.g., poly(dimethyldiallylammonium) P[DADMA]^+^, poly(benzylethyl) trimethylammonium P[VBTMA]^+^ and poly(1-vinyl 4-ethylimidazolium) P[C_2_VIm]^+^, as displayed by Scheme 1) and investigated how the tethered counterions influences the ER property under electric fields. The reason for the influence was explained by combining dielectric relaxation spectroscopy with activation energy measurement and density functional theory (DFT) calculation, which is an effective tool to understand the structure and dynamic of materials [19]. 

## 2. Materials and Methods

### 2.1. Chemicals

Diallyl dimethylammonium chloride ([DADMA]Cl, 60% wt% in water), p-vinylbenzyl trimethylammonium chloride ([VBTMA]Cl), 97%), bromoethane (99%), 1-vinylimidazole (99%), potassium hexafluorophosphate (KPF_6_, 99%). 2,2’-azobis(isobutyronitrile) (AIBN) was purchased from Sinopharm Chemical Reagent Co. Ltd., Shanghai, China. These chemicals were used as received except that AIBN was purified by recrystallization in methanol.

### 2.2. Synthesis of [C_2_VIm]Br

1-Ethyl-3-vinylimidazolium bromide ([C_2_VIm]Br) was synthesized by one-step procedure as follows: 1-Vinylimidazole (6.245 g, 0.065 mol) and bromoethane (8.253 g, 0.075 mol) were dissolved in 10 mL methanol. The resulting solution was stirred at 120 rpm and 35 °C in a three-necked flask under N_2_ flow. After 24 h, the precipitate was formed, filtered, washed with diethyl ether, and dried for 12 h at room temperature in vacuum to get [C_2_VIm]Br.

### 2.3. Synthesis of PILs

All PILs were synthesized via a two-step method including monomer polymerization and anion exchange.

Poly(diallyl dimethylammonium hexafluorophosphate) (P[DADMA][PF_6_]) was synthesized as follows: [DADMA]Cl (5 g, 60% aqueous solution) was dissolved in 10 mL DI water, AIBN (0.06 g) was added as initiator. The resulting solution was reacted for 12 h at 55 °C under stirring at 150 rpm and N_2_ flow. Then, acetone was added into the reacted solution and P[DADMA]Cl was precipitated. The precipitate was further washed by acetone several times to remove residual [DADMA]Cl. After that, the obtained P[DADMA]Cl was dissolved in water and added into KPF_6_ aqueous solution (10%, 20 mL) to form precipitate. The precipitate was filtered, washed by water several times, and tested by silver nitrate aqueous solution to clarify whether chloride ions were entirely removed after washing. Finally, the precipitate was dehydrated at 70 °C in vacuum to get P[DADMA][PF_6_].

Poly(p-vinylbenzyl trimethylammonium hexafluorophosphate) (P[VBTMA][PF_6_]) was synthesized as follows: [VBTMA]Cl (3 g) and AIBN (0.06 g) were dissolved in 25 mL ethanol and then the solution was reacted at 70 °C under 150 rpm and N_2_ flow. After reaction for 12 h, acetone was added and P[VBTMA]Cl was precipitated. The precipitate was further washed by acetone several times to remove residual [VBTMA]Cl. After that, the obtained P[VBTMA]Cl was dissolved in water and mixed with KPF_6_ aqueous solution (10%, 20 mL) to form precipitate. The precipitate was filtered, washed by water several times, and tested by silver nitrate aqueous solution to clarify whether chloride ions were entirely removed after washing. Finally, the precipitate was dehydrated at 70 °C in vacuum to get P[VBTMA][PF_6_]. 

Poly(1-ethyl-4-vinylimidazolium hexafluorophosphate) (P[C_2_VIm][PF_6_]) was synthesized as follows: [C_2_VIm]Br (3 g) and AIBN (0.06 g) were dissolved in 30 mL trichloromethane and then the solution was reacted at 70 °C under 150 rpm and N_2_ flow. After reaction for 12 h, acetone was added and P[C_2_VIm]Br was precipitated. The precipitate was further washed by acetone several times to remove residual [C_2_VIm]Br. After that, the obtained P[C_2_VIm]Br was dissolved in water and mixed with KPF_6_ aqueous solution (10%, 20 mL) to form precipitate. The precipitate was filtered, washed by water several times, and tested by silver nitrate aqueous solution to clarify whether bromide ions were entirely removed after washing. Finally, the precipitate was dehydrated at 70 °C in vacuum to get P[C_2_VIm][PF_6_].

### 2.4. Preparation of ERFs

First, three PILs were milled and sieved into particles of 5–15 μm. Then, the density of particles was measured via the method reported in our previous paper [12]. Simply, the particles were added into the pycnometer (5.0 mL) containing silicone oil and the pycnometer was placed in an ultrasonic cleaning bath and connected to a vacuum pump. After ultrasonication under reduced pressure for 30 s to remove air in the particles, the pycnometer was filled with additional oil and the density was measured. Finally, the particles were further dehydrated for 2 days at 80 °C in vacuum and mixed with 50 cSt silicone oil to get ERFs with particle volume fraction of 20 vol%. The volume of PIL particles was calculated by the ratio of mass to the measured density of particles.

### 2.5. Characterization and Measurements

The chemical group of PILs was determined by the Fourier transform infrared spectra on JASCO (Tokyo, Japan) FT/IR-470 Plus Fourier transform infrared spectroscopy. The molecular structure of PILs was determined by Bruker (Billerica, MA, United States) DPX-400 ^1^H nuclear magnetic resonance (^1^H NMR) spectrometer at 400 MHz with DMSO-*d*_6_ as solvent. The glass transition temperature (T_g_) of PILs was estimated by DSC Q200 differential scanning calorimeter (DSC) within 0–300 °C at 10 °C/min heating and cooling rate. T_g_ was exacted from the midpoint of the transition region in the second scanning. The micromorphology of PILs was determined by XENOCS (Sassenage, France) Xeuss2.0 wide-angle X-ray scattering (WAXS) spectrometer at 0.6 mA and 50 kV. 

The ER property of ERFs was tested on Thermal-Haake RS600 rheometer equipped with a 35 mm parallel plate system within 20–120 °C. The gap between plates was 1.0 mm. The testing details were similar to our previous report [13]. First, the ERFs were filled into the gap and pre-sheared for 60 s at 300 s^−1^ to remove structure history. Then, the electric field was applied and remained for 30 s to form a balanced chain structure. Finally, the flow curves of the shear stress vs. shear rate were measured by the controlled shear rate mode within 0.1–1000 s^−1^. 

The dielectric spectroscopy of ERFs was measured on Agilent (Santa Clara, CA, USA) 4284A precision LCR meter equipped with 16452A liquid fixture within angular frequency range of 1.26 × 10^2^–6.28 × 10^6^ rad/s and temperature range of 25–120 °C.

## 3. Results and Discussion

The chemical structure of three PILs is shown in Scheme 1. Figure 1 shows ^1^H NMR spectra of three PILs. It is observed that the three PILs have completely polymerized and no sharp characteristic peaks corresponding to IL monomers have been observed except for two sharp peaks at 2.51 ppm and 3.34 ppm due to DMSO-*d*_6_ and H_2_O [18]. From the ^1^H NMR spectra of P[DADMA][PF_6_] in Figure 1A, it can be seen that a broad peak at 1.16–1.37 ppm corresponds to the hydrogen of backbone, a broad peak at 3.18 ppm corresponds to the hydrogen on the methyl group linked with the nitrogen atom, and a broad peak at 3.79 ppm corresponds to the hydrogen on the carbon heterocycle. From the ^1^H NMR spectra of P[VBTMA][PF_6_] in Figure 1B, it can be seen that a broad peak at 1.36–1.64 ppm corresponds to the hydrogen of backbone, two broad peaks at 6.50 ppm and 7.11 ppm correspond to the hydrogen on benzene ring, a broad peak at 4.34 ppm corresponds to the hydrogen on the methylene group attached to the benzene, and a broad peak at 2.90 ppm corresponds to the hydrogen on the methyl group of ammonium ions. From the ^1^H NMR spectra of P[C_2_VIm][PF_6_] in Figure 1C, it can be seen that a broad peak at 1.37–1.69 ppm corresponds to the hydrogen of backbone and the methyl group at the tail end, a broad peak at 4.12 ppm corresponds to the hydrogen on the methylene attached to the imidazole ring, and three broad peaks at 7.18 ppm, 7.79 ppm and 9.01 ppm corresponds to the hydrogens on the imidazole ring [20].

Figure 2 shows the FT-IR spectra of PILs. It is noted that three PILs possess the characteristic bands coming from both cation and anion parts. The position of P-F stretching vibration characteristic bands of PF_6_^−^ at 840 and 558 cm^−1^ is same for three PILs. But the position of characteristic bands of cation part is different. The characteristic bands of P[DADMA]^+^ appear at 3048 cm^−1^ (C-H stretching vibration), 2953 cm^−1^ (CH_2_ bending vibration), and 1473 cm^−1^ (C-N stretching vibration). The characteristic bands of P[VBTMA]^+^ appear at 3049 cm^−1^ (CH_3_ stretching vibration), 2927 cm^−1^ (CH_2_ stretching vibration), 1488 cm^−1^ (CH_2_ bending vibration), and 1611 cm^−1^ (the vibration of benzyl groups). The characteristic bands of P[C_2_VIm]^+^ appear at 3168 cm^−1^ (C-H stretching vibration), 2996 cm^−1^ (CH_2_ stretching vibration), 1452 cm^−1^ (CH_2_ bending vibration), 1625 cm^−1^ (C=N stretching vibration), and 1558 cm^−1^ and 1160 cm^−1^ (imidazole ring stretching vibration). In addition, after comparing with the FT-IR spectra of monomers (see Appendix A), we have not found the absorption bands corresponding to C=C in these PIL samples, supporting the successful polymerization of PILs and no monomer presence in resulting PIL samples. For example, unlike P[DADMA][PF_6_], the FT-IR spectra of [DADMA][PF_6_] show the stretching vibration band of C=C at 1640 cm^−1^, the stretching vibration band of C-H connected to C=C at 3095 cm^−1^, and the bending vibration bands at 997 and 964 cm^−1^. Similarly, unlike P[VBTMA][PF_6_], the FT-IR spectra of [VBTMA][PF_6_] also show the stretching vibration band of C=C at 1634 cm^−1^, the stretching vibration band of C-H connected to C=C at 3044 cm^−1^, and the bending vibration bands at 991 and 914 cm^−1^. Unlike P[C_2_VIm][PF_6_], the FT-IR spectra of [C_2_VIm][PF_6_] has not only the characteristic bands of [C_2_VIm]^+^ but also the stretching vibration band of C=C at 1661 cm^−1^, the stretching vibration band of C-H connected to C=C at 3010 cm^−1^, and the bending vibration bands of C-H connected to C=C at 964 and 919 cm^−1^ [21].

Different from the crystallized structure of conventional ionic solids or polymeric ionic coordination complexes [22,23], most of PILs are amorphous solids. Thereby, WAXS spectra have been frequently employed to analyze morphology of structure [24,25]. We also used WAXS to characterize the micromorphology of PILs as shown in Figure 3. Three WAXS peaks labelled as q_p_, q_i_, and q_b_ can be observed in P[C_2_VIm][PF_6_], while one peak labelled as q_p_ and one shoulder-like peak labelled as q_i_ are observed in P[DADMA][PF_6_] and P[VBTMA][PF_6_]. The WAXS peaks come from the correlation distances and there are often three WAXS peaks in many PILs [26]. The lowest q_b_ is assigned to the correlation distance of main-chain to main-chain (d_b_), which also reveals the microphase separation caused by the polar region and non-polar region due to the unique structure of polyelectrolytes. Obviously, there is microphase separation or ion aggregation in P[C_2_VIm][PF_6_], while the microphase separation or ion aggregation become weak in P[VBTMA][PF_6_] and P[DADMA][PF_6_]. This may be related to the presence of strong π-π interaction of C_2_VIm^+^ pedant groups. The middle q_i_ is assigned to the correlation distance between PF_6_^-^ anions and PF_6_^−^ anions (d_i_), which can be calculated by the Bragg function d_i_ = 2π/q_i_ [26]. The order of the value of d_i_ is d_i P[DADMA][PF6]_ (0.661 nm) > d_i P[VBTMA][PF6]_ (0.640 nm) > d_i P[C2VIm][PF6]_ (0.632 nm). Due to the same PF_6_^−^ as counterions, the d_i_ value should be influenced by the size of pedant groups. Obviously, the size of DADMA^+^ (radius = 0.308 nm) is biggest among three samples, which coincides with its biggest d_i_. The size of C_2_VIm^+^ (radius = 0.304 nm) is bigger than that of trimethylammonium (radius = 0.283 nm), but d_i_ P[C_2_VIm][PF_6_] is smaller than d_i_ P[VBTMA][PF_6_]. This is because C_2_VIm^+^ tend to be planar and the presence of P[C_2_VIm]^+^ aggregation. The d_p_ is assigned to the correlation distance of pedant group to pedant group (d_p_) and can be calculated by the Bragg function d_p_ = 2π/q_p_ [19]. The order of the value of d_p_ is d_p P[DADMA][PF6]_ (0.472 nm) ≥ d_p P[VBTMA][PF6]_ (0.470 nm) > d_p P[C2VIm][PF6]_ (0.426 nm). The smaller d_p_ of P[C_2_VIm][PF_6_] is once again because of the presence of P[C_2_VIm]^+^ aggregation.

T_g_ of three PILs are 187 °C, 240 °C, and 170 °C for P[DADMA][PF_6_], P[VBTMA][PF_6_], and P[C_2_VIm][PF_6_], respectively. At room temperature, they are glassy state and easily milled and sieved into particles. The shape of particles is irregular and the size is 5–15 μm. The densities are 1.35 g/cm^3^, 1.45 g/cm^3^, and 1.63 g/cm^3^ for P[DADMA][PF_6_], P[VBTMA][PF_6_], and P[C_2_VIm][PF_6_] particles, respectively. The ERFs are prepared by dispersing the particles in silicone oil with the same volume fraction. Silicone oil is insulating and it has no ER effect. So, it has no influence on the comparison of ER effect of different PILs.

Figure 4 shows the flow curves of shear stress vs. shear rate for the ERFs containing PIL particles in silicone oil. Without electric fields, the three ERFs are low viscous fluids with a similar viscosity of about 0.17 Pa·s at 1000 s^−1^ because the size and shape of three PIL particles are similar. With electric fields, the ERFs show an increase in shear stress and a yield stress like a plastic material. As the strength of electric fields elevates, the shear stress and yield stress enhance. However, the intensities of yield stress and shear stress are different among three PIL ERFs.

Figure 5 plots the static yield stress (*τ_s_*) and the electric field-induced increment of shear stress (∆*τ*) at different electric fields. *τ_s_* is obtained by extrapolating the pseudo plateau stress in low rate region to zero, which can characterize the solidification level or the magnitude of ER property at yield point. ∆*τ = τ*_E_
*– τ*_0_, where *τ*_E_ is the shear stress at electric field and *τ*_0_ is the shear stress at zero electric field, respectively, which can characterize the magnitude of ER property in flow region [27]. Here, we calculate ∆*τ* at 100 s^−1^ because the hydrodynamic force is high enough to compete with electric field-induced interparticle interaction. It is seen that *τ_s_* and ∆*τ* depends on the tethered counterions attached to backbone and varied in the order of P[DADMA][PF_6_] > P[VBTMA][PF_6_] > P[C_2_VIm][PF_6_]. Especially, *τ_s_* and ∆*τ* of P[C_2_VIm][PF_6_] ERFs are significantly lower than those of P[DADMA][PF_6_] and P[VBTMA][PF_6_].

As temperature increases, three ERFs also maintain obvious ER property as shown in Figure 6, but the magnitude of ER property is different among the three PILs. The order is still P[DADMA][PF_6_] > P[VBTMA][PF_6_] > P[C_2_VIm][PF_6_] with the increase of temperature. *τ_s_* and ∆*τ* of P[C_2_VIm][PF_6_] ERFs are still significantly lower than those of P[DADMA][PF_6_] and P[VBTMA][PF_6_] at different temperatures. Therefore, the above rheological results clearly show the tethered counterions attached to backbone have an effect on the ER property of PILs.

Because the ER property is associated with the interfacial polarization of ER particles in carrier liquid [28], to understand the reason of the influence of tethered countercations on ER property, we employed dielectric spectroscopy to analyze the polarization characteristic of PIL ERFs. Figure 7 shows the angular frequency (ω) dependence of dielectric constant (ε′) and loss (ε″) of three PIL ERFs at different temperatures. At room temperature, three ERFs display dielectric dispersion but no relaxation peak. At relatively high temperatures, the relaxation peak appears and the peak position moves towards high frequency as the temperature rises. According to the analysis in our previous report [29], this relaxation process is attributed to the interfacial polarization induced by the movement and accumulation of dissociated PF_6_^−^ at the interface between PIL particles and silicone oil. To get good ER property, it has required ER particles to have not only large interfacial polarizability but also suitable polarization rate because ERFs are usually work under the simultaneous stimuli of electric and shearing fields [28]. The polarizability can be reflected by the dielectric strength (∆*ε′*= *ε′*_0_ − *ε′*_∞_, where *ε′*_0_ is the limit value of *ε′* below the relaxation frequency and *ε′*_∞_ is the limit value of *ε′* above the relaxation frequency), the polarization rate can be reflected by the dielectric relaxation time (*λ* = 1/*ω*_max_, *ω*_max_ is the angular frequency at ε’’ peak position). The polarization in phase with the exciting electrical field is available for ER effect. Even if the ERFs are subjected to a DC electric field, the *λ* values of particle polarization has been proposed to locate in or near a suitable range of 1.6 × 10^−3^ − 1.6 × 10^−6^ s because too fast or too low polarization rate is easy to cause interparticle repulsion or insufficient interaction under shear field. As *λ* decreases within 1.6 × 10^−3^ − 1.6 × 10^−6^ s and large ∆*ε′* is achieved, a strong ER property can be obtained [28]. To obtain the values of *λ* and ∆*ε′*, we fit the data in Figure 7 by the solutions of *ε*’ and *ε*’’ of the dielectric relaxation function below (Equation (1)) [30].
(1)$$\;\;\varepsilon^{*}\left(\omega \right)=\varepsilon^{\prime }+i\varepsilon^{\prime \prime }=\varepsilon_{\infty }^{’}+\frac{∆\varepsilon^{\prime }}{1+{\left(i\omega \lambda \right)}^{\alpha }}+{\left(i\frac{\sigma }{\varepsilon_{0}^{\;}\omega }\right)}^{\beta }$$
where, $\varepsilon_{\infty }^{\;’}+\frac{∆\varepsilon^{\prime }}{1+{\left(i\omega \lambda \right)}^{\alpha }}$ is the interfacial or dipole polarization contribution and ${\left(i\frac{\sigma }{\varepsilon_{0}^{\;}\omega }\right)}^{\beta }$ is the charge diffusion contribution including the conduction of ions in carrier liquid and the polarization effect at electrode. ε_0_ is permittivity of free space, σ is DC conductivity, α is the Cole–Cole parameter, and β is a fractional exponent (0 ≤ β ≤ 1). The solutions of ε’ and *ε*’’ as follows:(2)$$\varepsilon^{\prime }=\varepsilon_{\infty }^{\;}+∆\varepsilon^{’\left(\frac{1+{\left(\omega \lambda \right)}^{\alpha }\cos\left(\frac{\pi \alpha }{2}\right)}{1+2{\left(\omega \lambda \right)}^{\alpha }\cos\left(\frac{\pi \alpha }{2}\right)+{\left(\omega \lambda \right)}^{2\alpha }}\right)}+\left(\frac{\sigma }{\varepsilon_{0}\omega }\right){\;}^{\beta }\cos\left(\frac{\beta \pi }{2}\right)$$
(3)$$\varepsilon^{\prime \prime }=∆\varepsilon ’\left(\frac{{\left(\omega \lambda \right)}^{\alpha }\sin\left(\frac{\pi \alpha }{2}\right)}{1+2{\left(\omega \lambda \right)}^{\alpha }\cos\left(\frac{\pi \alpha }{2}\right)+{\left(\omega \lambda \right)}^{2\alpha }}\right)+\left(\frac{\sigma }{\varepsilon_{0}\omega }\right){\;}^{\beta }sin\left(\frac{\beta \pi }{2}\right)$$

The solid lines in Figure 8 show that Equations (2) and (3) well fit the data of *ε*’ and *ε*’’ The obtained dielectric parameters at room temperature are listed in Table 1. It is seen that the values of *λ* of three PILs are not located in the desired range of 1.6 × 10^−3^ − 1.6 × 10^−6^ s, but they depend on the type of tethered ions. The order of *λ* is *λ*_P[DADMA][PF6]_ < *λ*_P[VBTMA][PF6]_ < *λ*_P[C2VIm][PF6]_, while ∆*ε′*_P[DADMA][PF6]_ > ∆*ε′*_P[VBTMA][PF6]_ > ∆*ε′*_P[C2VIm][PF6]_. The orders well agree with the change order of ER property in Figure 4, Figure 5 and Figure 6. Therefore, the influence of tethered ions on ER property should be associated with the differences in the interfacial polarization rate and polarizability of PILs. 

It is known that the interfacial polarization depends on the conductivity of particles. As displayed in Table 1, the change of ∆*ε′* and *λ* corresponds to the conductivity (*σ_p_*) order of PIL particles. In three PILs, the conductivity originates from the motion of untethered PF_6_^−^. Thus, according to *σ = nqμ* (where *n* is the number density of mobile ions, *q* is the elementary charge, and *μ* is the ion mobility) [31], the number and mobility of free PF_6_^−^ are the key to the conductivity.

The number density ratio of total PF_6_^−^ can be approximately estimated to be *n*_P[DADMA][PF6]_:*n*_P[VBTMA][PF6]_:*n*_P[C2VIm][PF6]_ = 3.00:2.60:3.48 at same particle volume fraction by considering the molecular weight and particle density. It is seen that *n* of PF_6_^−^ in P[C_2_VIm][PF_6_] is the highest among three PILs, but its conductivity is the lowest. This reveals that the real number of free PF_6_^−^ contributing to conductivity is not in accordance with this because of different ion-pair complexation interaction energy (*E*_c_) [24]. The value of *E*_c_ can be calculated by DFT with Gaussian program by considering the different electronegativity of atoms and charge distribution of anion and cation in the real PIL structure as shown in Figure 8 [19,32]. Restrained electrostatic potential (RESP) atomic charges are obtained with Multiwfn program [33]. The values of *E*_c_ of P[DADMA][PF_6_], P[VBTMA][PF_6_] and P[C_2_VIm][PF_6_] are calculated to be 30.54, 36.40, and 57.32 kJ/mol by using integral equation formalism polarizable continuum model (IEFTCM) and considering the effect of dielectric constant of PILs. Obviously, the dissociation ability of PF_6_^−^ changes in the order of P[DADMA][PF_6_] > P[VBTMA][PF_6_] > P[C_2_VIm][PF_6_]. The value of *E*_c_ of P[C_2_VIm][PF_6_] is much higher than that of P[DADMA][PF_6_] or P[VBTMA][PF_6_]. This well agrees with the order of conductivity. Therefore, the difference in conductivity and polarization should be related to the number difference of free PF_6_^−^ in the three samples, which is further determined by ion-pair interaction energy with different tethered ions.

The second factor is the mobility of free PF_6_^-^. In glassy polyelectrolytes, the ion motion follows a hopping mode and the mobility is dominated by the charged number of ions, the vibration frequency of ions, the distance between two nearest neighbor hopping sites, and the activation energy barrier (*E*_a_) [34,35]. Because the same PF_6_^−^ ions act as mobile ions in these three PILs, the charged number of ions and the vibration frequency of ions are same. The distance between two nearest neighbor sites is approximately identical with the correlation distance of PF_6_^−^-PF_6_^−^ (d_i_) i.e., d_i P[DADMA][PF6]_ (0.661 nm) > d_i P[VBTMA][PF6]_ (0.640 nm) > d_i P[C2VIm][PF6]_ (0.632 nm) according to the WAXS result. The longer the distance is, the higher ion mobility is achieved. The order of d_i_ values agrees with the order of the conductivity and relaxation time as a function of tethered counterions as shown in Table 1. The value of *E*_a_ of ion motion within the glassy PIL particles can be calculated by the Arrhenius equation, *λ*^−1^ ∝ exp(−*E*_a_/*RT*), where *λ*^−1^ the reciprocal of relaxation time, *R* the universal gas constant and *T* the temperature in Kelvin [36]. Figure 9 plots the relation of *λ*^−1^ and 1000/T. It is seen that the values of *E*_a_ depend on the type of tethered ions and the change order is *E*_a P[DADMA][PF6]_ (75.91 kJ/mol) < *E*_a P[VBTMA][PF6]_ (84.53 kJ/mol) < *E*_a P[C2VIm][PF6]_ (104.04 kJ/mol). The lower *E*_a_ is, the higher ion mobility is achieved. The order of values of *E*_a_ agrees with the orders of *σ*_p_ and *λ* as a function of tethered ions as shown in Table 1. Therefore, the difference in conductivity and polarization among three samples should be also related to the difference in the mobility of free PF_6_^−^. Deeply, *E*_a_ is contributed by two parts including ion-pair interaction energy (*E*_c_) and elastic potential energy (*E*_el_) of PIL matrix, i.e.,$\;E_{a}=E_{c}+E_{\mathrm{el}}=E_{c}+\gamma G_{\infty }4\pi r_{1}^{3}/3$, where *G_∞_* is the high-frequency shear modulus of PILs and *γ* is constant that is usually smaller than 1 [37]. *E*_c_ has been calculated by DFT. Thus, according to the experimental values of *E*_a_, we can calculate the values of *E*_el_ are 45.37, 48.13 and 46.72 kJ/mol for P[DADMA][PF_6_], P[VBTMA][PF_6_] and P[C_2_VIm][PF_6_]. It is noted that the values of *E*_el_ for three samples are very close, which indicates that the difference of *E*_a_ should be related to ion-pair interaction energy.

By the above analysis, it can clarify the reason for the difference in the ionic conductivity and interfacial polarization among the PILs with different tethered counterions is mainly associated with ion-pair interaction energy that is resulting in different number and mobility of mobile ions. P[DADMA]^+^ as tethered ions has the smallest ion-pair interaction energy with mobile ions, which can result in the highest ionic conductivity and the fastest interfacial polarization rate for its strongest ER property.

## 4. Conclusions

The influence of tethered countercations on the electric polarization and ER property of PIL particles in insulating carrier liquid has been explored by synthesizing three PILs containing the same mobile anions but different tethered countercations. The P[DADMA]^+^ PILs show the strongest ER property, while P[C_2_VIm]^+^ PILs show the weakest ER property. This different ER property is mainly related to the ion-pair interaction energy between tethered ions and mobile ions that is affecting ionic conductivity and ion motion-induced interfacial polarization of PIL particles. The PILs with tethered countercations having small ion-pair interaction energy with mobile ions can result in high ionic conductivity and relatively fast interfacial polarization rate for strong ER property. This conclusion provides further insight to guide ER material design on molecular level.

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
