# Peer review of "Influence of Tethered Ions on Electric Polarization and Electrorheological Property of Polymerized Ionic Liquids"

_molecules, 2020, doi:10.3390/molecules25122896_

Round 1
Reviewer 1 Report
The authors have studied the effect of cation substitution in PIL on the ER properties of PIL. The work has been nicely carried using different experimental and theoretical methods. I don't have any major comments to add except that the literature review in the introduction can be more exhaustive covering theoretical studies on ionic liquids and similar solutes:
E Bodo Molecules 2020, 25(6), 1432
Similarly, the usage of theoretical tools Gaussian, Multiwfn is missing due citation.
Reviewer 2 Report
The authors present a study on the behavior of electrorheological fluids obtained from polymerized ionic liquids. Electrorheological fluids are interesting stimuli-responsive fluids that find applications in materials for shock absorption, vibration isolation, crude oil transportation, among others. The aim was to disclose structure-property relationships using different polymer backbones to better understand the electrorheological properties. The authors identified the reason for different ionic conductivities interfacial polarization among the PILs with different tethered counterions as being mainly associated with ion-pair interaction energy, which results in different number and mobility of mobile ions.
This is an interesting work and fits within the scope of MOLECULES, however, I recommend some revisions before the manuscript molecules-832654, can be accepted for publication.
- In the introduction, the information about electrorheological fluids doesn’t illustrate their potential applications. The authors have already published some works in with ERF and PILs (ex: reference 17), that should be emphasized in the introduction.
- In the ERF preparation, the authors should describe briefly how the density of particles
was measured. - The ERFs are prepared by mixing PILs with mixed with 50 cSt silicone oil. These preparations are said to yield ERF in 20 vol%. Being PILs solid particles that are milled, how can this preparation be in volume instead of weight?
- The rheological measurements are the focus of much of this work. Experimental detail should be provided.
- According to the accepted NMR NOMENCLATURE AND CONVENTIONS (IUPAC -Pure Appl. Chem., Vol. 73, No. 11, pp. 1795–1818, 2001), the graphical presentation of spectra should show frequency increasing to the left and positive intensity increasing upwards. Therefore, the authors should correct the chemical shift scale in all the 1H NMR spectra.
- In the 1H NMR spectra (fig. 1), being that the intended products are polymers, the respective signals are naturally broad, they should however be perceived. The intensity of the peaks should be incremented in order to see more than the solvents clearly.
- If the authors had presented the NMR signals with higher intensity, it would be clear that at least for P[C2VIm][PF6], there are still monomer signals (5 and 5.5 ppm). Moreover, in the same spectrum, proton 2 shouldn’t have the same chemical shift as proton 1.
- In the IR spectra (fig. 2) the monomers should also appear to facilitate the comparison, since contrary to what is stated, P[C2VIm][PF6] still has monomer present.
- In the discussion of WAXS data, could be helpful to compare ion volumes also.
- Shouldn’t equation 1 include the electrode polarization (EP) term? Shouldn’t the reference be 17, instead of 19?
Additionally:
- In line 75 where says “Bthyl-3-vinylimidazolium bromide” should say “Ethyl-3-vinylimidazolium bromide”.
- In line 101 where says “Poly(1-bthyl-4-vinylpyridinium hexafluorophosphate” should say “Poly(1-ethyl-4-vinylpyridinium hexafluorophosphate”.
- In line 204 where says “complete with” should say “compete with”. The conclusions should be rephrased.
- The manuscript should be revised by an English native to improve readability.
Reviewer 3 Report
The paper “Influence of Tethered Ions on Electric Polarization and Electrorheological Property of Polymerized Ionic Liquids” by Fang He, Bo Wang, Jia Zhao, Xiaopeng Zhao, Jianbo Yin describes phenomena of some interest. Polymer ionic liquids were studied according to the authors. When you look as unbiased reader to the paper you could say that the materials described are are ionic solids. The non characteristic WAXS spectra are not easily to be interpreted in an unambiguous way since less characteristic peaks were observed. The rheological measurements in Fig. 4 were performed with the addition of silicone oil. The influence of this ingredient is a little bit nebulous. The results in Fig. 5 and Fig. 6 are not very significant. The most important results are those in Fig. 7. Equation 1 is used for the simulation of the findings. Literature 19 is cited, but in looking in this paper this equation is not presented. In principle, eq. 1 is basic physics which describes polarisation contribution in phase and out of phase with the exciting electrical field strength. More comments would be helpful for the reader.
Generally, it must be stated that the paper has to be considerably improved before publication.
Round 2
Reviewer 3 Report
The revised paper “Influence of Tethered Ions on Electric Polarization and Electrorheological Property of Polymerized Ionic Liquids” by Fang He, Bo Wang, Jia Zhao, Xiaopeng Zhao, Jianbo Yin shows some deficiencies. An important problem is the title with “Ionic liquids”. Generally, liquids are “liquid” near room condition. The salt NaCl is also a liquid above 801°C, but it is not designed as ionic liquid. Now, the authors write in the introduction “solid polyelectrolyte” which is more convenient.
The new literature cited in respect to eq. (1) cannot be controlled!
The authors should more thoroughly consider referee comments I:
“When you look as unbiased reader to the paper you could say that the materials described are are ionic solids. The non characteristic WAXS spectra are not easily to be interpreted in an unambiguous way since less characteristic peaks were observed. The rheological measurements in Fig. 4 were performed with the addition of silicone oil. The influence of this ingredient is a little bit nebulous. The results in Fig. 5 and Fig. 6 are not very significant. The most important results are those in Fig. 7. Equation 1 is used for the simulation of the findings. Literature 19 is cited, but in looking in this paper this equation is not presented. In principle, eq. 1 is basic physics which describes polarisation contribution in phase and out of phase with the exciting electrical field strength. More comments would be helpful for the reader.
Generally, it must be stated that the paper has to be considerably improved before publication.”